# Improvement of the Technology of Precision Forging of Connecting Rod-Type Forgings in a Multiple System, in the Aspect of the Possibilities of Process Robotization by Means of Numerical Modeling

**DOI:** 10.3390/ma17051087

**Published:** 2024-02-27

**Authors:** Marek Hawryluk, Łukasz Dudkiewicz, Sławomir Polak, Artur Barełkowski, Adrian Miżejewski, Tatiana Szymańska

**Affiliations:** 1Department of Metal Forming, Welding and Metrology, Wroclaw University of Science and Technology, Lukasiewicza Street 5, 50-370 Wroclaw, Poland; lukasz.dudkiewicz@pwr.edu.pl (Ł.D.); slawomir.polak@pwr.edu.pl (S.P.); artur.barelkowski@pwr.edu.pl (A.B.); 2Schraner Polska, Lotnicza Street 21G, 99-100 Łęczyca, Poland; amizejewski@schraner.pl (A.M.); tszymanska@schraner.pl (T.S.)

**Keywords:** hammer precision forging, forging FEM modeling, robotization, technology improvement

## Abstract

The study refers to the application of numerical modeling for the improvement of the currently realized precision forging technology performed on a hammer to produce connecting rod forgings in a triple system through the development of an additional rolling pass to be used before the roughing operation as well as preparation of the charge to be held by the robot’s grippers in order to implement future process robotization. The studies included an analysis of the present forging technology together with the dimension–shape requirements for the forgings, which constituted the basis for the construction and development of a thermo-mechanical numerical model as well as the design of the tool construction with the consideration of the additional rolling pass with the use of the calculation package Forge 3.0 NxT. The following stage of research was the realization of multi-variant numerical simulations of the newly developed forging process with the consideration of robotization, as a result of which the following were obtained: proper filling of the tool impressions (including the roller’s impression) by the deformed material, the temperature distributions for the forging and the tools as well as plastic deformations (considering the thermally activated phenomena), changes in the grain size as well as the forging force and energy courses. The obtained results were verified under industrial conditions and correlated with respect to the forgings obtained in the technology applied so far. The achieved results of technological tests confirmed that the changes introduced into the tool construction and the preform geometry reduced the diameter, and thus also the volume, of the charge as well as provided a possibility of implementing robotization and automatization of the forging process in the future. The obtained results showed that the introduction of an additional rolling blank resulted in a reduction in forging forces and energy by 30% while reducing the hammer blow by one. Attempts to implement robotization into the process were successful and did not adversely affect the geometry or quality of forgings, increasing production efficiency.

## 1. Introduction

Connecting rods are elements that join the drive systems in many machines and devices, including petrol chainsaws used, e.g., in the woodworking industry. They constitute an important safety element of the drive system and for this reason, they have to be characterized by high dimension–shape precision, quality, and performance properties [1]. In large lot production, for forgings of the connecting rod type, hot precision forging is applied in open or closed dies, usually on crank and screw presses, as well as hydraulic hammers, often in multiple systems, in order to improve the production efficiency with the preservation of the high product quality [2]. The obtained forgings are subjected to flash trimming as well as some additional procedures (cooling, thermo-chemical treatment, shot peening, finishing treatment through machining, defectoscopic tests, and final inspection). The process of producing connecting rod-type forgings, especially in multiple systems, still constitutes a big challenge and a not entirely solved problem, as, in the die forging processes, a huge role is played by the development of a proper tool construction and working impression geometry as well as selection of the correct technological parameters of the process, which translates to technology efficiency and product quality, with the consideration of tool durability. This is of significant importance in the case when the forgings are required to have an additionally specified structure and hardness, which are obtained as a result of both the forging process and, often, a proper thermal treatment [3]. The most important factors affecting these aspects are the technological parameters, the shape and quality of the tools as well as the number of operations, the geometry of the preform and the slug forging, and also the thermal parameters affecting the tribological conditions [4,5]. Although the forging technology is relatively well-known, the proper preparation of the forgings, especially ones of a complicated shape, will fulfill the precision and quality requirements demanded from the constructors and technologists, as well as operators, to possess extensive knowledge and experience [6]. A certain aid as well as solution in the increase in stabilization and efficiency of production is the introduction of automatization and robotization of the forging processes [7], however, it is an additional challenge, requiring time and financial resources. The literature provides few applications of effective automatization and robotization of die forging processes, which, however, are dedicated to specific types of forgings, and each one requires an individual approach [8,9]. This creates the need to design the technology from scratch, as it requires a redesign of the tools and consideration of the role of the robots’ grippers in the process [10]. In the case of hot die forging, one should also consider the abrasive wear of the tools’ working impressions during production, which causes material losses, which, in turn, increase the forging volume at the expense of the forging material, which makes the gripping of the forging by the robot significantly more difficult, and this should also be considered in the design of automated forging processes. In the case of forging process automatization, one should especially take into account not only the vibration but also the large elastic deflections of the whole tool sets, which requires the application of compensation in the gripper joints/their docking, or the application of other solutions [11]. Such solutions coupled by means of machine communication enable control of the slide as well as the time of the hold-up for the required time [12,13]. This is also connected with the implementation of additional sensors and detectors, elements of electronics, as well as the application of specialized software, and also the necessity of performing numerous tests and trials [14]. The attempts at introducing automatization and robotization of hot forging processes, in the case of using at least a part of the outdated machinery park, are even more difficult [15,16]. At the same time, at every stage of the technological line, there is a potential risk of the occurrence of a defect/problem lowering the quality of the products or causing unfulfillment of the geometrical or microstructural requirements (improper structure after forging, inappropriate hardness, etc. [17]. Also, many times, the cause of the errors identified at a given stage is an improperly developed technology, also in the aspect of robotization, and/or it not being followed at the earlier stages of the process [18]. As we know, a properly elaborated industrial process of plastic treatment requires the realization of numerous experiments and tests, which is connected with huge costs and a lot of devoted time [19], and still, the most important stage of design and optimization is the final verification of the developed process under industrial condition [20,21]. For this reason, at present, for the analysis, optimization, or elaboration of new technology, engineering programs based on CAD/CAM/CAE are used, which are aided by IT tools [22,23,24], as well as the very popular numerical modeling, which often constitutes an independent tool [25,26]. The simultaneous application of many methods and techniques aiding the design, simulation, and production enables a global and complex approach to the given problem [27,28]. The available literature provides many studies and articles referring to the use of numerical modeling techniques for the analysis and optimization of plastic-forming processes. In such a case, numerical modeling based on FEM/FVM [29] is mostly applied for the determination of the optimal shape and dimensions of the slug forging, as well as the material flow and filling of the impression, and also the temperature field, both in the forging and the tools, or for the determination of the deformation distribution, and even the grain size after the process [30]. The studies [31,32] apply FEM for a complex analysis of the forging process in order to improve the analyzed technology. In turn, in the works [33,34], FEM was used for an analysis of the process of flash trimming. The currently used calculation packages are equipped with more and more functions enabling an even better and more thorough analysis of the metal forming processes [35,36,37], making it possible, e.g., to analyze the wear of the dies and forging punches (Forge, QFORM, Simufact, Deform) [38,39]. More and more often, in FEM [40,41], special functions for flaw detection are applied [42]. The application of such functions by the user makes it possible to significantly shorten the implementation time of a new project and limit the errors, e.g., during the design of new instrumentation taking into account the gripping by the robots’ grippers [43]. Although numerical modeling and IT tools [44] significantly change the role and scope of the experiment to the virtual dimension, the real experiment remains, on the one hand, the best and most necessary verification, and on the other hand, it is the most expensive and time-consuming stage of design [45]. Nonetheless, numerical modeling has been and still is a very convenient, fast, and most commonly applied tool for the analysis and optimization of production processes, including the robotization and automatization of die forging processes [46]. What is more, a continuous expansion of such IT tools with new functions and capabilities makes it possible to approach the real experiment even closer. The additional support of the numerical modeling results with measurement techniques (3D scanning, etc.), as well as microstructure examinations, enables a complex analysis of the whole technology, as well as its improvement and development [47,48].

The aim of the research is to construct a proper numerical model of a hot forging process of producing a connecting rod forging in a triple system, in order to improve the current technology, especially in the aspect of introducing an additional rolling pass (to increase efficiency), as well as optimizing the technological parameters of the process and the possibilities of robotization of the precision forging process realized so far. 

## 2. Test Subject and Methodology

In the research, an analysis was performed of a 3-type forging elongated forgings-high (with a stem) with protrusions of a connecting rod type, produced in a triple system. Figure 1a presents the application of this forging product. A CAD model of a single forging is shown in Figure 1b. A dimensioned forging is presented in Figure 1c.

The current process of producing a connecting rod forging is realized in 2 forging operations: roughing and finishing forging. The charge material is a bar, with a diameter of 16 mm and 280 mm long, made of carburizing steel 13CrMo4-5 (1.7335). The number of details forged in one cycle: 3. The tools (forging dies) are made of 1.2367 steel, and, after the thermal treatment, their hardness is at the level of 53–55 HRC. After being cut, the charge material is heated in an induction furnace, which is followed by forging on a hydraulic hammer with energy of 16 kJ. Whether the connecting rod forgings are properly made is determined by the following requirements: dimension–shape accuracy at the level of 0.1–0.3 mm, mean grain size of 5 according to ASTM, and hardness at the level of 170–200 HB, as well as lack of surface defects or laps; joggle at the level of 0.3 mm is accepted. Technological allowances for this type of connecting rods range from 0.5 mm to 1 mm per side. In the technology implemented so far, the process is realized manually. 

In order to achieve the set goal, the following methods and techniques, as well as measurement and testing tools, were used in the realization of the consecutive research and development studies:-a complex analysis of the forging process with the use of, e.g., a thermovision camera Flir 840 (Wilsonville, OR, USA) and a high-speed camera (Casio Pro Ex-F3, Casio, Tokyo, Japan), as well as a macroscopic analysis of the tools and the forging defects by means of a camera Cannon EOSx 60D (Cannon, Tokyo, Japan).-three-dimensional scanning Atos Core 135 (GOM, Braunschweig, Germany) structured light scanner, equipped with two 5MPix CCD cameras (GOM, Braunschweig, Germany) (resolution 2448 × 2050 pixels) with a measuring field of 135 × 100 mm, working distance of 170 mm, and physical point distance of 0.05 mm for a single scan.-based on the current specification sheet, development of CAD models of a ready forging as well a tool (additional rolling pass) with the consideration of the aspects of robotization (increased length of the charge through the use of the so-called “tickworm” enabling a double-sided grip by the manipulators’ grippers) by means of the program Catia V6R20 by Dassault, Paris, France.-based on the above information, a numerical model was developed and simulations of the improved technology of hot precision forging were made with the use of the calculation package of the QForm program (https://www.qform3d.com/ (accessed on 21 February 2024)) in order to determine the key parameters and physical quantities as well as identify the most important problems.-modeling of the trajectory of the robots’ movement (RobotSudion ABB 2020).-in order to verify the introduced change and improvements resulting mostly from the numerical modeling, a measurement of the forgings geometry was made in reference to a forging before the changes, as well as an analysis of the obtained microstructure and hardness of the produced forgings.-microstructural observations (for verification purposes) with the use of a light microscope Leica Dm6000N (Leica Microsystems, Tokyo, Japan). To that end, the die insert was incised along the shorter side to prepare samples for the tests. The grinding and polishing, in order to obtain traditional micro-sections, was conducted on a grinder-polisher Struers 350. For the etching, a picric acid solution was used.-hardness measurements made by means of a hardness tester LECO LC120 (LECO, St. Joseph, MO, USA);

Figure 2 presents a diagram in order to better understand the test process.

It should be emphasized that a thorough implementation of full robotization combined with automatization includes many more technological and technical aspects, which have not been discussed in this article (e.g., automatization of the loading and temperature segregation, determination of the local key devices, selection of robots, design of the manipulation grippers, assembly and replacement of tools, manner of their installation on the hammer as well as heating and lubrication of the tools in the forging process, etc.).

## 3. Results and Discussion

The research was divided into a few stages, among which the first three included an analysis of the technology implemented so far in the manual system, as well as the design and modeling of the forging process with the consideration of the aspects of robotization. In turn, the last stage referred to preliminary tests under industrial conditions for verification of the performed research.

### 3.1. Analysis of the Current Technology in a Manual System

Forgings characterized by an elongated shape with a big difference between the cross-sessions along the length, such as connecting rods, are usually made from a slug forging, in which the material is already preliminarily formed. Forgings can be produced in different ways. The most efficient method is the use of dedicated machines, i.e., forging rolling mills, in which the milling can take place by the method of transverse rolling or periodic rolling. However, in the case of no rolling mill, the forming process can happen directly on the hammer through open or semi-open die forging in the dedicated spot on the die. In the current process of forging a connecting rod, a multiple system is used, in which three forgings are made from a round bar. The forging process takes place in two operations. The first forging operation, i.e., roughing, consists of flattening a cylindrical preform placed on the diameter. This operation involves the highest pressures and material deformations. The second operation is finishing forging, as a result of which we obtain a shape close to that of the ready product. The lubricant is used as a mixture of graphite with water in proportion 1:16. The efficiency of the current technology is 2100 items per shift (700 leaves); the cycle time for the manual process is 19 s. Figure 3 shows the thermograms from the process for the charge, the tools, and the forging.

The temperature measured by the pyrometer of the heater was 1320–1340 °C. The temperature of the charge leaving the heater is about 10–20 °C higher than that measured when the charge is lying on the feeder and waiting to be received by the operator. In turn, the working temperature of the tools is 200–250 °C. The heated charge material is fed into the roughing pass, in which roughing forging takes place by way of two blows, and next, in the finishing pass, one blow is performed. The final shape of the product is obtained as a result of hot trimming of the forging. A certain problem is the cooling off of the tools during forging, which is caused by the small volume of the deformed material with respect to the mass of the dies and the insufficient energy provided to the tools as a result of deformation and its work being exchanged into heat. In one forging cycle, the tools are in contact for a maximum of 0.5 s, whereas during the remaining time, the tools cool down as a result of radiation, conduction, and, to a small extent, convection with the environment. This is an issue which, in the case of process automatization, should also be solved in order to ensure prolonged work of the robots, without the necessity of frequent heating of the tools. Preliminary plans include the use of fast induction heating [49]. 

To sum up the performed analyses referring to the current technology, we should state that the production of a forging directly from a bar causes the formation of excessive flash, which is disadvantageous both with respect to die durability and a large material loss. Additionally, forging in a manual system causes a lack of stability and repeatability in the process, a decreased efficiency, and also translates to a lowered quality of the forged items and the formation of forging defects. For this reason, in order to improve the current technology, changes and solutions were introduced consisting of the development and design of a new tool construction with the consideration of forging in a robotized system. One of the implemented solutions is the use of an additional rolling pass for a preliminary re-forming of the material and the use of a longer charge in order to ensure the so-called tickworm, enabling the grip and holding of the charge/forging by the manipulators’ grippers during the robotized forging process. To the current length of the input material in the form of a cylindrical bar, taking into account three times the length of a single forging together with the technological allowance, an allowance should be applied on both sides (about 20–30 mm each) for the so-called forceps, which will allow robots to hold the rod on both sides while forging. Additionally, a decision was made to reduce the charge diameter from 20 mm to 18 mm, which should translate to smaller material losses and lowered energy–force parameters. Introducing automatization and robotization of the currently realized technology should make it possible to solve the above problems and bring measurable benefits as well as improve the current forging process.

### 3.2. An Ideal Slug Forging and Rolling Pass

Before designing the rolling pass, we should first determine the ideal slug forging. To that end, we should calculate the section areas of the forging together with the flash in the particular areas. As the whole consists of three identical forgings, it is enough to calculate the section areas of one forging and next reflect it on the others. The assumed initial data are the die model, from which we cut out the body of the forging together with the flash, 10 mm wide, which is a sufficient value in the aspect of an easy-to-fill (during forging) shape of the connecting rod as well as a small height in reference to the width. On this basis, the particular section areas of the isolated body of the fogging were determined with respect to the length (Figure 4).

The diagram of the sections presented this way was smoothed so that it would be possible to use the correlated ideal slug forging for the preparation of the rolling pass (Figure 4c). With the data referring to the ideal slug forging, a rolling pass for a round bar, 14 mm in diameter, was designed. It was assumed that it would be a closed rolling pass, that is, of an elliptical shape. In such a case, according to the literature, the width of the pass b (the larger diameter of the ellipsis) should correspond to 3/2 of the height h (the smaller diameter of the ellipsis). The complete 3D model used to prepare the roller is shown in Figure 4d. Next, as a result of further research, a roller with a magazine was designed, which served to reduce the amount of energy needed for the deformation in the rolling pass. However, after the simulation, this solution was abandoned because there were no noticeable differences between the rollers in both versions, whereas the use of a magazine would involve unnecessary milling. After being connected to the lower and upper die (Figure 5), the roller obtained its final shape.

In order to avoid the flash moving between the dies, in the area of its possible occurrence, the gap between the dies, 1.6 mm high, was expanded up to the border of the locks. 

### 3.3. Modeling of the Forging Process with Rolling

The forging process of connecting rod-type forging on a robotized station will be realized with the application of an additional rolling operation, which was developed in order to reduce the diameter of the charge material from 20 mm to 18 mm. Special attention in the simulation was paid to the rolling operation, for which a dozen or so variants were created. During the deformation process, the tips of the bar were not deformed as they were to be held by the robots. The conditions assumed in the calculations of the forging process on the automatized station were exactly the same as those for manual forging, only the length of the bar was larger, and the shape of the die was adjusted in a way that would enable a non-collision placement of the robot’s grippers between the upper and lower die. Also, in the roughing operation, one blow of the hammer was applied. The conditions in the calculations were assumed for a process of forging in three-pass dies. The number of blows: RX—the rolling pass—one blow; 1X (the roughing pass) with the energy of 10.5 kJ—one blow; 2X (the finishing pass): 6.2 kJ—one blow. The forging temperature and the charge temperature was 1320 °C. The cycle time was 12 s divided into cooling 6 s + forging in three operations. The machine was a hydraulic hammer with 16 kJ of energy. The tool temperature was 250 °C. The lubrication was water with graphite. The heat exchange was an average of 10 kW/(m^2^·K). The simulation results for a Ø 18 mm diameter bar are presented in Figure 6 (models A and B).

For models A and B, we can see certain differences in the construction of the tools in the pre-roughing operation. Modification A has no bridge for the flash, whereas, in modification B, we can see the shape of a bridge (the color red denotes lack of contact with the die). Modification A has stronger friction forces on the flash, which can bring a more advantageous result in successive forging operations. The shape of the pass was previously filled, so a 1.2 mm opening was used, whereas, in modification B, a 1 mm opening was chosen (Figure 6b). The difference in the dimension of the forging measured in the direction of the hammer’s stroke is in favor of solution A, as it is 0.2 mm bigger (Figure 6a). The results of the simulation in the roughing operation show that, regardless of the shape of the used flash, the filling of the pass is fully realized for a 14 mm diameter bar. For this reason, further works were initiated in order to perfect the technology, with the final assumed solution from model A. The approximate forming forces and the forging energy are shown in Figure 7 for the pre-roughing operation. The energy was about 5 kJ and the forging force was maximally 250 tons.

A comparison was also made of the forces and energies of formation in the manual and automated forging technology in the roughing and finishing operation (Figure 8). In the roughing operation in the manual forging process, the hammer performs two blows, whereas in the case of adding the rolling operation and the application of a smaller diameter bar, i.e., 18 mm, we need one blow with the energy of about 10 kJ. 

The forming force can be lower in the case of the technology in three operations, and, for exemplary results, it equals about 500 tons, previously being over 800 tons. In turn, the energy needed to deform the detail in the finishing operation is lower in the simulation and equals 4.5 kJ, while in the case of manual forging, it is over 6 kJ. However, the force increased from 700 to 800 tons.

During the process, the temperature increases as a result of a change in the plastic deformation work into heat and, in the roughing operation, it equals about 945–1145 °C, which is presented in Figure 9. The plastic deformation in this operation equals maximally about 5—the highest takes place in the area of the changes in the cross-section.

The temperature at the end of the forging process after the finishing forging equals about 890–1100 °C (Figure 10). The highest temperature is on the side of the upper die. On the cross-section, we can see a big difference in the temperature field distribution.

The maximal plastic deformations increase only slightly. The deformation process runs properly. The formed lap occurs on the flash beyond the detail and does not affect the quality of the obtained products.

A detailed analysis was also made of the changes in the grain size during the forging in the newly elaborated robotized technology, in order to provide a possibility to model the microstructural changes during deformation and thus verify the usefulness of numerical modeling for such tasks connected with a microstructural analysis. The initial grain size was assumed at the level of 45 µm, based on the performed microscopic tests and analyses. In the QForm program, the mean grain size (its diameter in µm) is defined as follows:(1)$$d_{\mu m}=1000\sqrt{{\;10^{-}}^{\frac{ASTM+2.95}{3.32}}}$$
where ASTM can be read from the table below [38] (Table 1).

In turn, Figure 11 shows the grain size distributions in the final phase of the rolling operation, with a full pocket right after the forging process and 2 s after the forging (this is the time when the forging is replaced into the successive pass of the roughing forging operation).

Based on the presented grain size distributions in the final phase of the process, we can observe that the largest mean grain diameters equaling 38–41 µm are localized in the areas of the forging with the highest volume, that is where the material was the least deformed. In areas of larger deformations, the grain is much smaller (Figure 8b). The case is similar for the roughing forging operation, where the deformation is at the level of 2–4, which means the grain size, as a result of microstructure reconstruction (dynamic recrystallization), is at a similar level (Figure 12).

Figure 13 shows the results of the forging simulation for the fishing operation, which included the distributions of deformation, temperature changes, and grain size right after the forging and 3 min after the end of the process.

We can notice that, as a result of the dynamic processes, the recrystallization did not take place in the analyzed time frame in the whole volume of the forging (Figure 12c), with respect to the deformation distribution for the forging in the final forging phase. In turn, we can see that, in the case of a longer time (Figure 13d,e), it has no significant effect on the grain growth. Moreover, the obtained modeling results for the time of 180 s after forging can be compared to the results of the microstructural tests for the forging after forging on the robotized station, because, as we can see, throughout a longer period of time, no significant changes in the microstructure can be observed. The mean grain size obtained from the numerical modeling equals about 35–40 µm (6 according to ASTM).

The presented results with the grain size, plastic deformation, and temperature distributions demonstrate that, in reference to the results for the roughing operation, the changes in the grain size and plastic deformations are not that big, which results from the fact that the finishing operation is in a sense an operation of calibration. And so, the forging does not undergo large deformations, which makes the effect of the time elongation on the gain size and the plastic deformations small.

## 4. Trials under Industrial Conditions and Numerical Modeling Verification

During the robotization of the forging process, it was decided that, from the moment of placing the heated material onto the die to the moment of it being moved to the hot trimming operation, the formed element would be held on both sides by the robots’ grippers, which would be moving synchronically during all the forging operations (Figure 14a). This would ensure stable support in two points, which would prevent the material from sliding out. The three robots assigned for the work would be performing the following tasks.

Robot R1 supports the heated bar, which has been moved out of the induction heater, and places it above the lower die in such a way that the other end can be gripped by robot R2. Both robots manipulate the forging during the forging process, after which robot R1 releases its grip and the further manipulation takes place only by robot R2, which intercepts the forgings connected by the flash and next, places them in the trimming press tool through the side window. Figure 14b presents the preliminary forging tests performed on the robotized station.

In order to verify the properness of the forgings produced on the robotized seat, measurements of selected forged elements were made. In the first place, 3D scanning was carried out of the test batch of the connecting rod forgings collected from the forging process before the trimming. Presented below are representative scanning results. A cloud of points in the form of a triangle grid was obtained, which, after the measurement data equalization, was analyzed by means of the GOM software (version 2019). Figure 15 shows exemplary 3D scanning results in the form of a colored map of deviations for one leaf randomly selected from the whole series.

Analyzing the data shown in Figure 15, we can notice deformation of the forging’s leaf, which is formed as a result of the forging process. We can see that, in the central part, which was used to equalize the data, the shape deviation is the smallest. In the case of the seat on the left, a slight deformation of the forging occurs. In the case of the forging forged on the right seat, we can see deformation of the external part of the forging at the level of −0.91 mm. Such a bend should not significantly affect the dimension–shape precision of single forgings. Nonetheless, detailed measurements of six selected leaves were made: two from the beginning of the process, two from the middle, and two from the end of the technological trials, and the obtained results are presented in Table 2.

As we can notice, the obtained measurement for the selected geometrical characteristics (Dm—the diameter of the small mesh; Dd—the diameter of the big mesh; Tm—the thickness of the small mesh; Tg—the thickness of the big mesh; and D—the joggle), are basically within the assumed dimension tolerances, which shows that the forging process in the developed robotized system is appropriate with respect to the dimension–shape accuracy of the forgings. Only for the first two leaves, the thickness results of the small mesh demonstrated that it was slightly higher, yet for the consecutive randomly selected elements, we can see that the process has stabilized, and all the geometrical features are in the tolerance field.

Additionally, in order to verify the performed preliminary forging tests in the robotized system, the initial material, i.e., steel 13CrMo4-5 (1.7335), was examined. The steel is chrome–molybdenum steel, containing 0.70–1.15% Cr and 0.4–0.6% Mo, making it one of the most noble and durable materials. Chromium steel has a relatively low carbon content compared to other related grades that are also used for high-temperature applications. The performed microscopy tests were carried out on a light microscope for microstructure verification. Figure 16 shows the results for the microstructure of the material as-delivered steel 1.7335.

The microstructure of steel 16MnCr5 is a typical low-carbon ferritic–pearlitic microstructure with visible fine Fe_3_C precipitates in the form of coagulated particles, not bound in the form of pearlite. It has a fine-grained structure. The measured hardness for this material as delivered equaled about 218 HV. In turn, for a randomly selected forging (from the middle part of the technological trials), metallographic tests as well as microstructural and grain size analyses were performed (in order to confront the results with those of numerical modeling). From the forging, three samples for each test were cut out. Figure 17 shows the areas of the shearing line for the samples for micro-section preparation, on the surface of which microstructural examinations were made as well as grain size evaluation and hardness measurements HV1 were performed.

Figure 18 presents the microstructure test results for selected areas from the forging together with the measurement of the grain size. In turn, Table 3 shows the collective results of the microstructural analysis for a connecting rod forging. The microstructure was revealed by way of etching with a 2.5% Nital reagent and the microstructure observations were performed on the laser microscope Keyence VHX 6000 with magnifications of 100× and 200×. The determination of the grain sizes in the microstructures of the analyzed connecting rod was made by the secant method, with the use of the specialized software of the microscope.

The obtained test results referring to the microstructure confirm that introducing process robotization into the current technology has no negative effect on the microstructural changes, as the grain size, as well as the hardness and microstructure, are according to the requirements. In turn, referring to the grain size results obtained from the technological trials to FEM, we can state that the grain size is one grade lower (5 according to ASTM, that is about 65 µm) with respect to the modeling results (6 according to ASTM, that is 40 µm). On this basis, we can state that automatizing the current forging process realized in a multiple system through a transfer from a manual to an automatized process brings only advantageous aspects (dimension–shape precision and microstructure) as well as additionally ensures stability and repeatability of the production process. This said, before introducing the newly developed forging technology into series production, further tests and investigations should be carried out under industrial conditions.

## 5. Conclusions

The study presents the results referring to an improvement as well as the possibilities of introducing robotization into the currency-realized technology of precision forging on a hammer to produce connecting rod forgings with the use of mainly numerical modeling. Based on the analysis of the state of the art, it was proven that the available literature provides hardly any thorough studies or solutions for robotized forging seats; what is more, robotization cannot be treated as a universal solution. For this reason, the article has proposed a solution of optimizing the presently realized process of forging connecting rod forgings with the consideration of the aspects of robotization, e.g., by way of developing an additional rolling pass, reducing the diameter and mass of the charge material, and holding the charge/slug forging by robots’ grippers. The investigations included a complex analysis of the present forging technology to determine the key areas requiring improvement/changes and next designing a new tool construction and modeling the technology by means of the calculation package Forge 3.0 NxT through multi-variant numerical simulations of the newly developed forging process. The obtained results have been verified under industrial conditions and correlated with respect to the forgings obtained in the technology realized so far. The attempts at introducing robotization aimed mostly at making the present manual process stable and repeatable, which would increase the quality of the forgings and at the same time reduce the rejection rate as a result of eliminating the errors produced by the operator. The obtained results of technological tests have confirmed that the introduced and then implemented changes were proper and brought measurable benefits. Both the dimension–shape accuracy and the hardness and microstructure are according to the requirements. The additional rolling pass reduced the forging forces and energy by over 30% and decreased the number of hammer blows by one, with respect to the manual process, which should also translate to an increased hardness of the forging instrumentation. This said, in order to ultimately verify the elaborated solution, it is necessary to perform further examinations and tests including a much longer operation time. It should also be emphasized that the presented results point to a big potential of the use of numerical modeling methods, as it currently enables full analysis of the process (including simulation of the microstructure development) as well as improvement of the given forging process.

## Figures and Tables

**Figure 1 materials-17-01087-f001:**
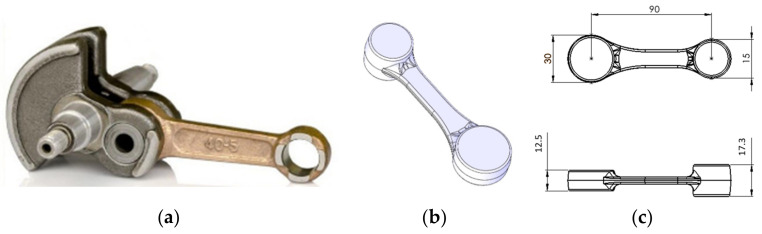
View of (**a**) a CAD model, (**b**) a photograph of the forgings after the consecutive stages of the manual production process, and (**c**) a single-dimensioned forging.

**Figure 2 materials-17-01087-f002:**
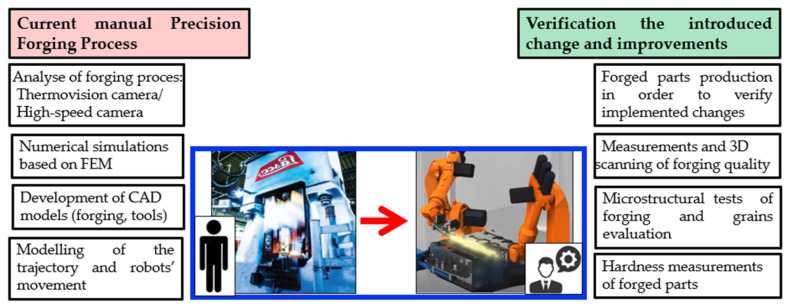
Diagram of the test process.

**Figure 3 materials-17-01087-f003:**
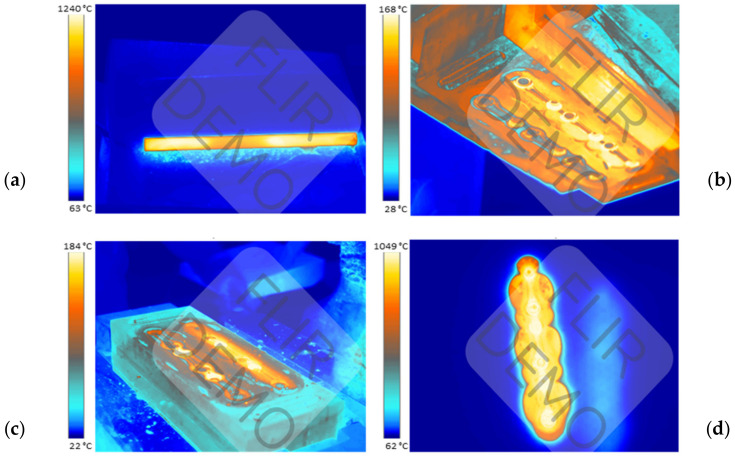
View with the temperature distribution during forging: (**a**) the charge lying on the feeder; (**b**) the temperature results for upper tools; (**c**) the results of lower tools; and (**d**) the temperature of forging.

**Figure 4 materials-17-01087-f004:**
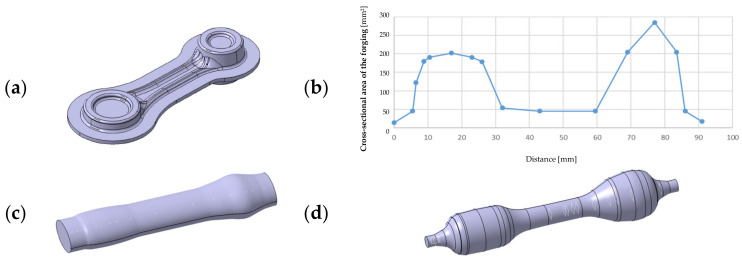
View of (**a**) a forging model with the predicted flash, (**b**) a diagram of the forging’s sections, (**c**) a correlated ideal slug forging, and (**d**) a 3D model used to prepare the roller.

**Figure 5 materials-17-01087-f005:**
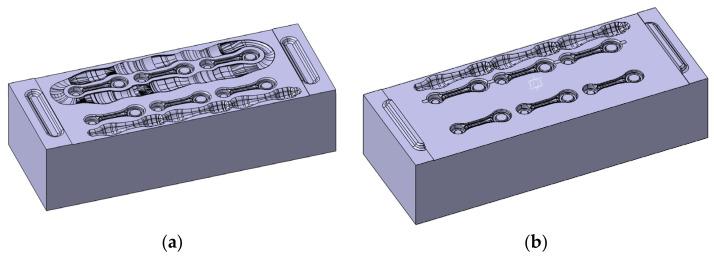
A 3D model of (**a**) the upper die with the roller and (**b**) the lower die with the roller.

**Figure 6 materials-17-01087-f006:**
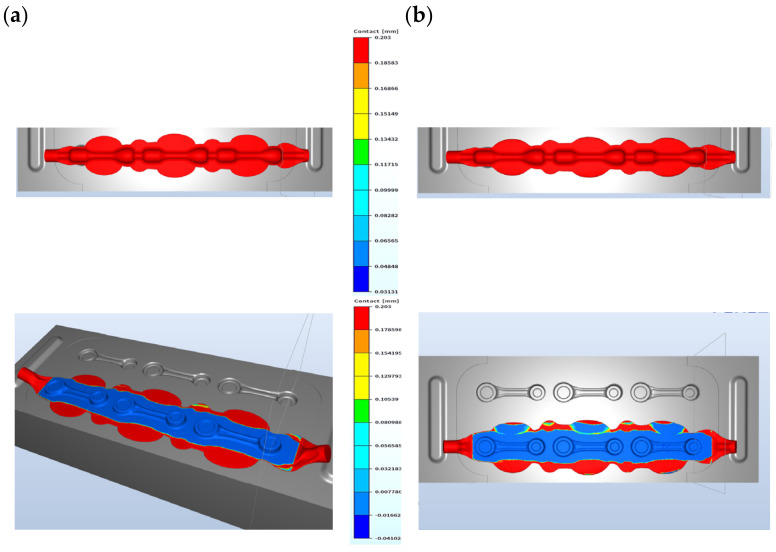
Contact with the tools in the pre-roughing operation, a 14 mm diameter bar: (**a**) model A—1.2 mm opening and (**b**) model B—1 mm opening.

**Figure 7 materials-17-01087-f007:**
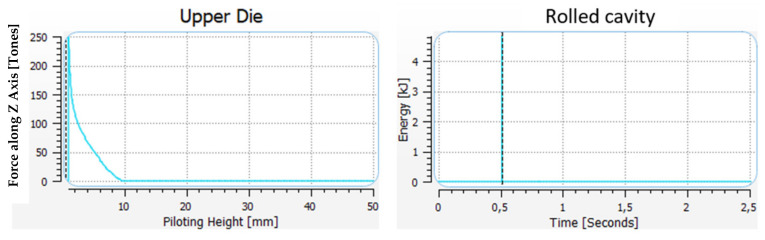
Force and energy of the blow—pre-roughing operation, variant B, a 1 mm opening.

**Figure 8 materials-17-01087-f008:**
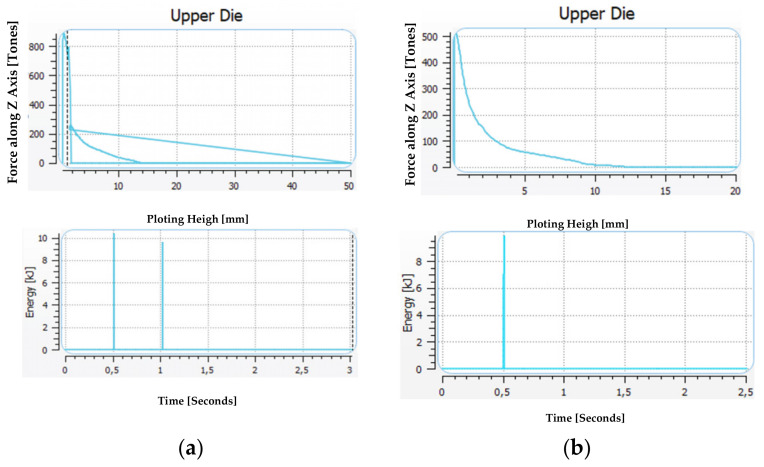
Comparison of the force–energy parameters for forging technology: (**a**) manually and (**b**) robotization.

**Figure 9 materials-17-01087-f009:**
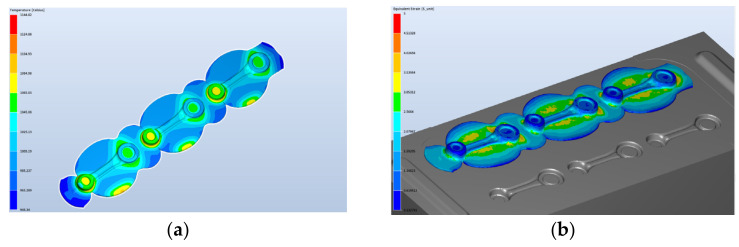
Results of the simulation after the roughing operation—distributions of (**a**) the temperature field and (**b**) the deformations.

**Figure 10 materials-17-01087-f010:**
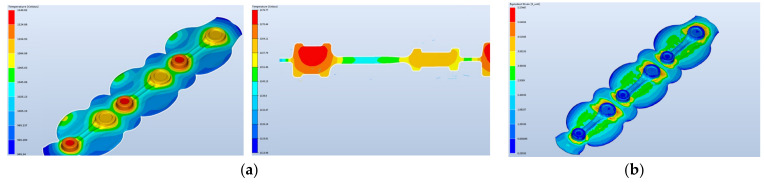
Results of the simulation after the finishing operation—distributions of (**a**) the temperature field and (**b**) the deformations.

**Figure 11 materials-17-01087-f011:**
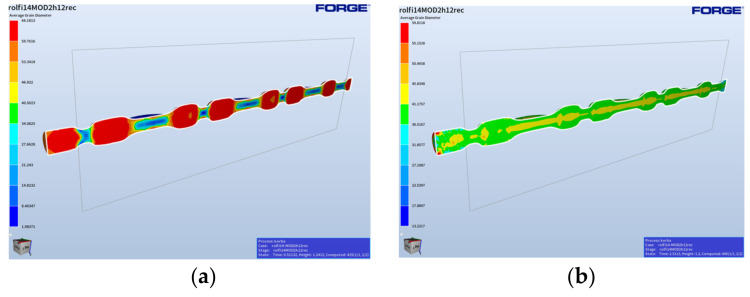
Results of the simulation for the rolling operation: (**a**) the final deformation phase and (**b**) 2 s after the rolling process.

**Figure 12 materials-17-01087-f012:**
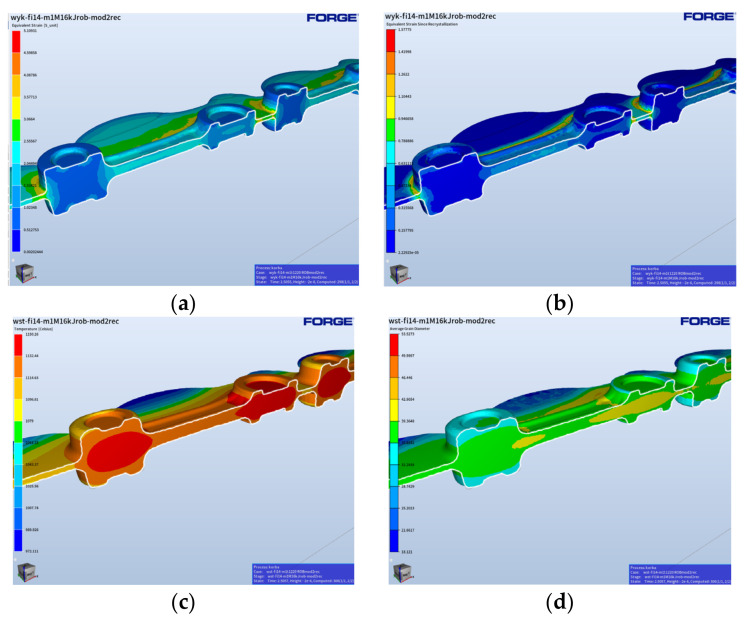
Results of the simulation for the roughing operation: (**a**) the total deformation distribution, (**b**) the deformation distribution only in the roughing forging (since recrystallization), (**c**) distributions of the temperature filed in the forging, and (**d**) grain size distributions.

**Figure 13 materials-17-01087-f013:**
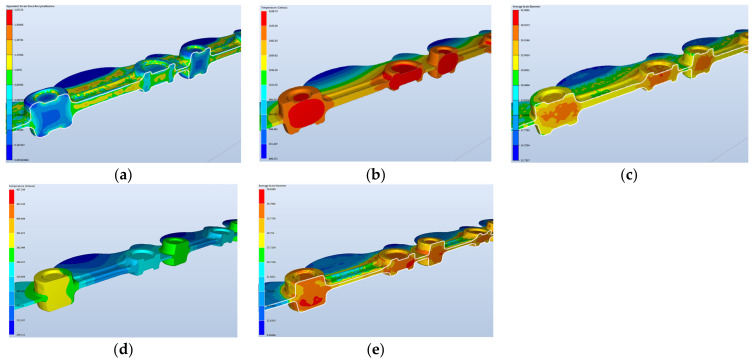
Results of the simulation for the finishing operation—the final phase of deformation: (**a**) deformation distributions, (**b**) temperature field distributions, (**c**) grain size distributions for this case, (**d**) temperature changes 3 min after the forging, and (**e**) grain size distributions for an analogical process.

**Figure 14 materials-17-01087-f014:**
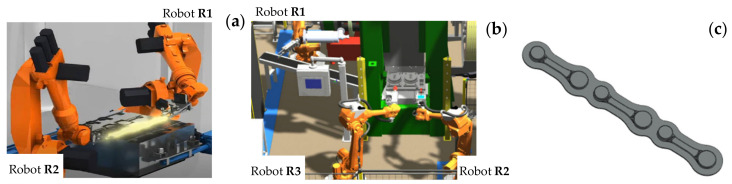
View of (**a**) an idea of the simultaneous work of both robots, (**b**) the simulation on the robotized station in the RobotStudio program, and (**c**) the CAD model of forged elements, the so-called “leaves”, with single forgings.

**Figure 15 materials-17-01087-f015:**
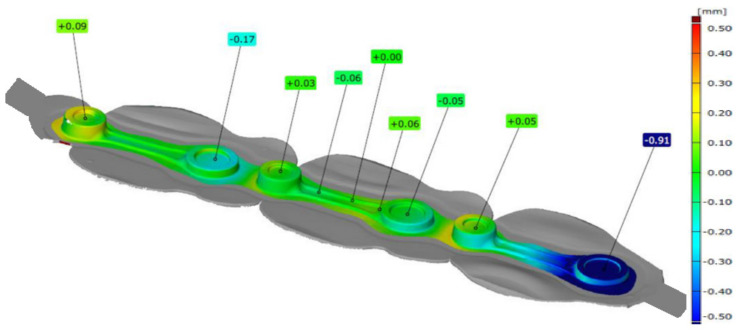
Results of 3D scanning of the K22 forging.

**Figure 16 materials-17-01087-f016:**
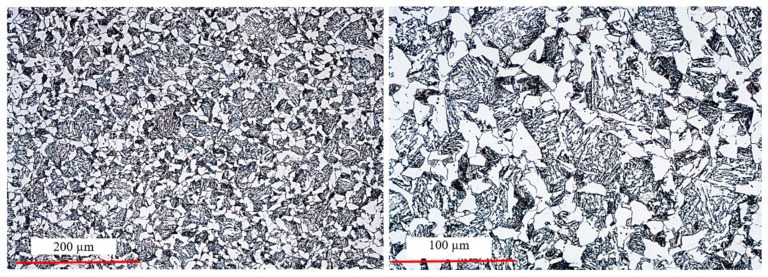
Microstructure of steel 1.7335—initial material for a connecting rod forging.

**Figure 17 materials-17-01087-f017:**
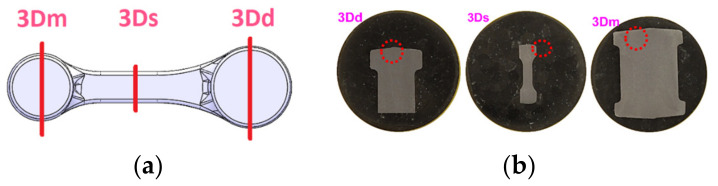
Photographs of (**a**) the areas where the examined connecting rod was cut for the metallographic test samples for the connecting rod and (**b**) the appearance of the microsections from the cut-out samples and the mounted ones.

**Figure 18 materials-17-01087-f018:**
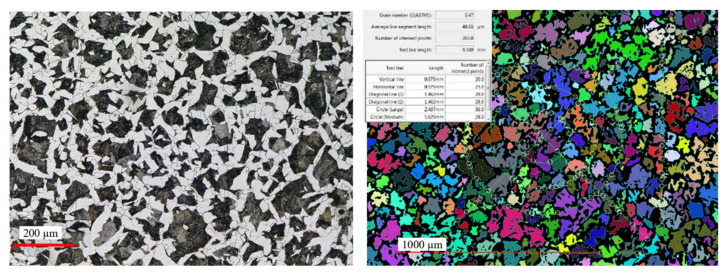
Results of the microstructural tests for selected areas from a randomly collected forging of a connecting rod: samples 3-Dd, 3-Ds, 3-Dm.

**Table 1 materials-17-01087-t001:** Grain size referred to ASTM.

ASTM	15	14	13	12	11	10	9	8	7	6	5	4	3	2	1	0
**Grain size (μm)**	2	3	4	6	8	11	16	22	32	45	63	90	127	180	254	360

**Table 2 materials-17-01087-t002:** Results of the measurements of selected geometrical features for 6 randomly chosen forged elements (for example, 1B denotes 1 forged leaf, 2—the number of a single connecting rod forging).

The First Attempt					
	Dd (mm) 30 ± 0.03	Dm (mm) 15 ± 0.2	Tm (mm) 12.5 ± 0.15	Td (mm) 17.3 ± 0.15	Deviation (mm) Max 0.3
1A	30.02	15.08	12.71	17.22	0.099
1B	29.99	15.07	12.72	17.24	0.048
1C	29.98	15.08	12.74	17.14	0.144
2A	29.99	15.04	12.68	17.21	0.064
2B	29.98	15.06	12.70	17.25	0.080
2C	30.03	15.07	12.72	17.20	0.124
The second attempt					
1A’	30.01	15.07	12.58	17.22	0.079
1B’	30.02	15.04	12.63	17.26	0.089
1C’	29.98	15.04	12.58	17.23	0.162
2A’	29.99	15.05	12.58	17.23	0.029
2B’	29.98	15.06	12.67	17.28	0.060
2C’	30.02	14.84	12.61	10.25	0.127
The last attempt					
3A”	29.99	15.05	12.60	17.19	0.028
3B”	29.98	15.06	12.61	17.25	0.075
3C”	30.03	15.04	12.59	17.20	0.180
3A”	30.01	15.07	12.58	17.20	0.029
3B”	30.02	15.06	15.62	17.27	0.082
3C”	29.98	15.06	12.60	17.23	0.039

**Table 3 materials-17-01087-t003:** Results of microscopic tests and hardness measurements.

Sample	Microstructure	Grain of Former Austenite PN-EN ISO 643:2020 [49]	Hardness HB	Decarburization Zone
3-Dd	ferrite + pearlite, ferrite present on grain boundaries	4.5	175	None
3-Ds	ferrite on grain boundaries + pearlite + bainite	5	175	None
3-Dm	bainite + ferrite on grain boundaries + traces of pearlite	5	190	None

## Data Availability

Data are contained within the article.

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
