# Peer review of "Improvement of the Technology of Precision Forging of Connecting Rod-Type Forgings in a Multiple System, in the Aspect of the Possibilities of Process Robotization by Means of Numerical Modeling"

_materials, 2024, doi:10.3390/ma17051087_

Round 1

Reviewer 1 Report

Comments and Suggestions for Authors

Interesting paper to read, a few comments for authors to consider:

1. In the abstract section, please can the authors quantify your research results?

2. Compare with the other techniques used for forging, what is new in your research?

3. A state of the art literature review of forging technology development would help the reader to understand the importance of the work that the authors proposed in this paper.

4. Figure 1c - the thickness of the connecting rod and the dimensions of the neck of the rod on each side should be given in the CAD model.

5. pg4, A diagram to show how the test has been set up would be essential to understand the test process.

6. pg 5 - please check Figure 1b mentioned in section 3.1 is the correct one.

7. Please explain Figure 2 in detail

8. pg 6 What is the technological allowance? Can you give a value for allowance?

9. Figure 6 what is Rolfi14mod5h1 referred to?

10. Table 1. ASTM Number against grain size- how these number defined in ASTM standard?

11. It would be useful to annotate Figure 13 so RobotR2 can be located.

12. The annotation in Figure 14 need to be enlarged to improve the readability.

13. Table 2 - please give SI units to some parameters.

14. pg14, what is the temperature used to obtain pearlite?

15. Section 5 could be conclusion rather than summary.

Comments on the Quality of English Language

English is fine, minor editing will be required.

Author Response

Dear Reviewer,

Thank you for your insightful review and valuable comments. We have referred to them in detail below, taking into account any necessary changes in the text of the manuscript.

We have provided detailed answers to the questions in a separate file.

regards,

Reviewer 2 Report

Comments and Suggestions for Authors

The paper explores the optimization and potential integration of robotization into precision forging processes, specifically focusing on the production of connecting rod forgings. Through numerical modeling, the study redesigns the forging process, incorporating features like additional rolling passes and reduced charge material diameter to enhance dimension-shape accuracy, increase stability, and reduce rejection rates. It presents simulation results illustrating temperature distributions, grain size analyses, and deformation patterns for various forging operations. Industrial trials are conducted to validate the proposed process, with measurements and analyses confirming the accuracy and quality of the produced forgings. Microstructural analysis compares experimental results with numerical predictions to assess simulation accuracy. Ultimately, the paper advocates for the benefits of incorporating numerical modeling and robotization into forging processes to improve product quality, process stability, and efficiency. I consider that the article can be published once minor corrections are made.

1.      The article presents promising results and is generally well-written. However, there is a lack of transparency regarding the numerical modeling aspect, which is crucial for understanding the optimization process. While the authors utilized the Forge 3.0 NxT package for determining optimal parameters in robotization, they should provide more insight into the physical and thermodynamic principles underlying the simulated processes. Specifically, it would be beneficial to expand on the models employed by Forge 3.0 NxT to simulate the forging process. Since Forge 3.0 is commercial software, clarifying the physical models it utilizes would enhance the comprehensibility of the study.  
If the essence of the contribution lies solely in the modeling aspect, the article may be perceived as limited in value beyond showcasing the authors' proficiency in software utilization. This could potentially hinder its usefulness, particularly for readers without access to the specific modeling software discussed. To enhance the article's relevance and accessibility, it would be beneficial for the authors to provide broader insights and explanations beyond the technicalities of the software, offering a more comprehensive understanding of the underlying principles and implications of the research findings.

 2.      Specify the units of the  geometrical parameters in table 2 

Author Response

(The authors gave the same response as above.)

Reviewer 3 Report

Comments and Suggestions for Authors

The manuscript needs revision. For the details see the attached file.

Comments on the Quality of English Language

The English style is acceptable in general. However, some corrections are necessary: forgotten text parts in Polish language can be found, as well as strange rules for writing capital letters. For more details see the attached file, too.  

Author Response

(The authors gave the same response as above.)

Round 2

Reviewer 1 Report

Comments and Suggestions for Authors

The authors have made effort to revise the paper and the quality of the paper has been improved.